

# Improved Methods for Signal Processing in Measurements of Elemental Mercury Vapor by Tekran® 2537A and 2537B Instruments

Jesse L. Ambrose[1]

[1]College of Engineering and Physical Sciences, University of New Hampshire, Durham, 03824, USA

*Correspondence to*: Jesse L. Ambrose (Jesse.Ambrose@unh.edu)

**Abstract.** Atmospheric Hg measurements are commonly carried out using Tekran® Instruments Corporation's model 2537 Hg vapor analyzers, which employ gold amalgamation pre-concentration sampling and detection by thermal desorption/atomic fluorescence spectrometry. A generally overlooked and poorly characterized source of analytical uncertainty in those measurements is the method by which the raw Hg atomic fluorescence signal is processed. Here I describe new software-

based methods for processing the raw signal from the Tekran® 2537 instruments, and I evaluate the performances of those methods together with the standard Tekran® internal signal processing method. For test datasets from two Tekran® instruments (one 2537A and one 2537B), I estimate that signal processing uncertainties in Hg loadings determined with the Tekran® method are within ±[6% + 0.94 pg]. I demonstrate that the Tekran® method produces significant low biases (≥5%) not only at low Hg sample loadings (<5 pg) but also at tropospheric background concentrations of gaseous elemental mercury (GEM) and total

mercury (THg) (~1 to 2 ng/m³) under typical operating conditions (sample loadings of 5−10 pg). Signal processing uncertainties associated with the Tekran® method therefore represent a significant unaccounted for addition to the overall ~10 to 15% uncertainty previously estimated for Tekran®-based GEM and THg measurements. In comparison, estimated signal processing uncertainties associated with the new methods described herein are low, ranging from within ±0.053 pg, when the Hg thermal desorption peaks are defined manually, to within ±[2% + 0.080 pg] when peak definition is automated. Mercury

limits of detection decreased by ~31 to 88% when the new methods were used in place of the Tekran® method. I recommend that signal processing uncertainties be quantified in future applications of the Tekran® 2537 instruments.

## 1 Introduction

Gold amalgamation pre-concentration, followed by thermal desorption (TD) in Ar carrier gas and detection via atomic fluorescence spectrometry (AFS) is a commonly used method for quantifying atmospheric elemental mercury vapor, $Hg^0(g)$

(hereafter referred to as gaseous elemental mercury, GEM) (Schroeder et al., 1995; Gustin and Jaffe, 2010; Pandy et al., 2011). Coupled with various sample capture and pre-treatment methods, the above measurement scheme is also used for quantitative analysis of atmospheric gaseous oxidized mercury (GOM) (Stratton and Lindberg, 1995; Landis et al., 2002; Lyman et al., 2007), total gaseous mercury (TGM ≡ GEM + GOM) (Ambrose et al., 2015), atmospheric particle-bound mercury (PBM) (Landis et al., 2002), atmospheric total mercury (THg ≡ GEM + GOM + PBM) (Jaffe et al., 2005), and total aqueous Hg

(USEPA, 2002). Most AFS-based atmospheric Hg measurements employ Tekran® Instruments Corporation's model 2537 Hg



vapor analyzers (versions A and B; hereafter referred to collectively as 'the Tekran® analyzer') (Schroeder et al., 1995; Landis et al., 2002; Tekran Corporation, 2006, 2007).

Several previous reports identified analytical uncertainties as critical in limiting scientific understanding of environmental Hg cycling (Jaffe and Gustin, 2010; Pirrone et al., 2013; Jaffe et al., 2014). An often overlooked source of analytical uncertainty

is the method by which the raw Hg atomic fluorescence (AF) signal is processed by the Tekran® analyzer. Although most researchers rely on the Tekran® analyzer's embedded software to automatically integrate the Hg TD peaks, Swartzendruber et al. (2009a) and Slemr et al. (2016) demonstrated that the accuracy and precision of the Tekran® peak integration method can significantly decline at low Hg loadings.

To better characterize analytical uncertainty associated with atmospheric Hg measurements made with the Tekran® analyzer,

and in an attempt to improve upon existing measurement methods for atmospheric Hg, I developed new software-based methods for offline processing of the raw Hg AF signal from the Tekran® analyzer. Here I describe the key features of the new signal processing methods and characterize their performance, together with that of the standard Tekran® signal processing method.

## 2 Experimental

Using National Instruments™ LabVIEW software (version 12.0), I developed a virtual instrument (VI) for offline processing of the serial data output ("RAWDUMP" format) from Tekran® models 2537A and 2537B mercury vapor analyzers. The VI is characterized and validated using data collected as part of the Nitrogen, Oxidants, Mercury, and Aerosol Distributions, Sources, and Sinks campaign (https://www.eol.ucar.edu/field_projects/nomadss) using a Tekran® 2537A instrument and a 2537B instrument incorporated in the University of Washington's Detector for Oxidized Hg Species (Swartzendruber et al.,

2009b; Lyman and Jaffe, 2011; Ambrose et al., 2013, 2015). The instrument configurations are described in Ambrose et al. (2015). Instrument operating parameters are given in Table S1 in the Supporting Information. The most significant difference between the two instruments tested is that hardware in the 2537B instrument was modified to improve the instrument's signal-to-noise ratio (Ambrose et al., 2013).

All linear regression equations reported herein were calculated by the Bisquare method (unweighted) in LabVIEW. All stated

uncertainties represent 95% confidence intervals, unless otherwise specified.

### 2.1 Virtual Instrument Design Overview

### 2.1.1 Operation

The VI's Hg atomic fluorescence signal processing method parses the Tekran® analyzer's serial data output ("RAWDUMP" format; see Sect. S1 in the Supporting Information) using text delimiters associated with each component (Figs. S1 and S2 in

the Supporting Information).





At the start of an analysis, the VI computes the overall mean baseline standard deviation, $\sigma_{bl}$, for all samples in the data file to be analyzed. The baseline standard deviation for each sample is first calculated as the 10 point (1 s) running mean of the baseline measurements made from 1 s after the start of the AF signal recording interval to 2 s prior to the approximate start of the Au trap desorption cycle (*i.e.*, the "BL time" specified by the operator in step 2 below). The value of $\sigma_{bl}$ is then calculated

5     from the values for all individual samples.

For each sample analysis cycle, the VI carries out the following operations:

1.     The "Raw data" string (Fig. S1 in the Supporting Information) is converted to an array of 10 Hz AF signal values.

2.     The timestamp, cycle type flag, and trap identity are extracted from the "Final data" string (Fig. S1 in the Supporting Information).

3.     An *x-y* plot is generated from the data array created in step 1.

4.     Unless the VI is set to automatically define the Hg thermal desorption peaks, the user is prompted to identify the start time, $t_{start}$, of the TD peak. This selection is accomplished by first manually setting the placement of a cursor on the *x-y* plot generated in step 3 and then using a control to extract the associated coordinates.

15     Alternatively, values of $t_{start}$ can be set to values defined automatically during the initialization procedure (see below). The mean baseline voltage at the start of the TD peak is calculated over the interval from $t_{start}$ to $t_{start} - 9$ ds ($n = 10$ data points).

5.     The coordinates of the Hg TD peak maximum are identified automatically from the data array created in step 1. In my experience, the Tekran® analyzer's baseline generally decreases (slopes downward) across the Hg TD peak. I parameterized the VI for such a condition by identifying the TD peak maximum as the largest AF signal value recorded after $t_{start}$.

6.     A preliminary TD peak height value is calculated from the mean baseline voltage at the start of the peak (step 4) and the maximum voltage (step 5). In cases when the preliminary peak height is a negative number, the VI sets the value to $\sigma_{bl}$. (See Sect. S3 in the Supporting Information for further details.)

7.     Based on user-defined settings on the VI's front panel, the TD peak end coordinates are either selected manually (as for the peak start time in step 4) or calculated automatically from the peak maximum in step 5 and the initialization parameters (as described below).

8.     The baseline beneath the Hg TD peak is calculated. For this purpose, the baseline at the beginning of the peak is defined by the coordinates of $t_{start}$ (step 4) and the preceding nine AF signal values. The baseline at peak end is similarly defined by the peak end coordinates in step 7 and the trailing nine AF signal values. The baseline coordinates beneath the peak are calculated by linear regression ($n = 20$ data points).

9.     The baseline standard deviation at the Hg TD peak is estimated from the mean residual of the regression in step 8 (see Sect. S4 in the Supporting Information for further details). I denote this value as "$\sigma_{bl, fit}$" to differentiate it from $\sigma_{bl}$ defined above.



10. The baseline-slope-corrected Hg TD peak height is calculated from the peak maximum voltage (step 5) and the calculated baseline voltage at the time of the peak maximum (step 8).

At the end of an analysis, an output data file is created, which contains, among other parameters, the baseline-slope-corrected Hg TD peak height (step 10) for each sample in the data file that was processed.

Hereafter I refer to three different configurations of the VI: "$VI_{m,m}$" when the peak start and end times are both identified manually; "$VI_{m,a}$" when only the peak start time is identified manually, and "$VI_{a,a}$" when the peak start and end times are both identified automatically. I abbreviate the Tekran® peak integration method as "the Tekran® method".

### 2.1.2 Initialization

The $VI_{m,a}$ and $VI_{a,a}$ methods are initialized by fitting eq. 1 to the Hg thermal desorption peaks recorded for a pair of calibration gas analysis cycles (*i.e.*, one A SPAN and one B SPAN; example fits are shown in Fig. 1 and in Fig. S3 in the Supporting Information).

$$S(t) = (A \times e^{b \times t}) + S_{offset} \qquad (1)$$

Here, $S(t)$ is the time-dependent 10 Hz Hg fluorescence signal along the tail of the Hg TD peak, $A$ is the peak amplitude (which is approximately equal to the peak height, $H$), $b$ is the decay constant, $S_{offset}$ is the baseline offset (*i.e.*, the difference between $lim[S(t)]_{t \to \infty}$ and zero), and $t$ is expressed in units of deciseconds (ds). The fit includes 150 $S(t)$ values, starting with the peak maximum, $S_{max}$ (Fig. 1 and Fig. S4 in the Supporting Information). The values of $S_{offset}$ and $A$ are constrained to the baseline minimum, $S_{min}$, and $S_{max} - S_{offset}$, respectively. (For the data shown in Fig. 1, the values of $S_{offset}$ and $A$ are approximately equal to 70 and 170 mV, respectively.)

The two decay constants (one for each Au trap) are the key parameters derived from the initialization procedure (see Sect. 2.1.3). I estimate the uncertainty in each $b$ value from the linear sum of two terms: (1) the relative difference between unity and the slope of a linear regression fit to a plot of the calculated versus measured Hg TD peak decays (Fig. 1b and Fig. S4b in the Supporting Information); and (2) the 95% confidence interval (CI) in the slope of the fit. For the data shown in Fig. 1, the uncertainty in $b$ is estimated to be 9%.

During initialization, peak start times, $t_{start}$, are also calculated for the pair of calibration gas analysis cycles. For this purpose, the VI defines $t_{start}$ as the first 10 Hz Hg fluorescence signal value in a series of seven consecutively increasing signal values. If the operator chooses (via a control on the VI's front panel) to automatically define the Hg TD peak start times, the values of $t_{start}$ calculated during initialization are assigned (paired by Au trap identity) to all samples in the data file to be processed.

### 2.1.3 Automatic Determination of the Hg Thermal Desorption Peak End Time

It is necessary to define each Hg thermal desorption peak's end time, $t_{end}$, within the interval during which the Tekran® analyzer's Hg atomic fluorescence signal is recorded (38.9 s in Fig. 1a and Fig. S4a in the Supporting Information). Therefore, for each sample the VI defines $t_{end}$ as the time at which $S(t)$ decays to a value equal to, or less than, a fraction, $f$, of the amplitude, $A_{SPAN}$ (derived as in Sect. 2.1.2), determined from a calibration gas analysis cycle on the same Au trap:



$$S(t_{end}) \leq (f \times A_{SPAN}) + S_{offset} \tag{2}$$

Substituting the right-hand side of eq. 2 for $S(t)$ in eq. 1, and solving for $t$ ($= t_{end}$) yields an analytical expression for the upper-bound value of $t_{end}$:

$$t_{end} = ln\left(f \times \frac{A_{SPAN}}{A_i}\right) \times b^{-1} \cong ln\left(f \times \frac{A_{SPAN}}{H_i}\right) \times b^{-1} \tag{3}$$

Here, $A_i$ and $H_i$ are the TD peak amplitude and the initial peak height (from step 6 in Sect. 2.1.1), respectively, of the sample for which the baseline-slope-corrected peak height is to be quantified. The partial equality in eq. 3 reflects that facts that $A_{SPAN}/A_i \cong H_{SPAN}/H_i$ and $A_{SPAN} \cong H_{SPAN}$. The Au-trap-dependent value of the decay constant, $b$, is derived as in Sect. 2.1.2. The value of $f$ is chosen such that $t_{end} \geq 0$ at the smallest expected value of $H_i$, which the VI estimates from $\sigma_{bl}$ (Sect. 2.1.1). The upper-bound value of $f$ is estimated separately for each Au trap by solving eq. 3 for $f$, with $t_{end} = 0$ and $H_i = \sigma_{bl}$:

$$f = \left(\frac{\sigma_{bl}}{A_{SPAN}}\right) \tag{4}$$

The Au trap-dependent $f$ values are then averaged prior to application in eq. 3. I estimate the uncertainty in $f$ to be twice the standard deviation in the mean for the two Au traps. I estimate uncertainty in the value of $H_i$ to be equal to twice the baseline standard deviation calculated in step 9 of Sect. 2.1.1. I estimate uncertainty in the automatically derived value of $t_{end}$, $\delta t_{end}$, by propagating estimated uncertainties in $b$, $f$, and $H_i$ through eq. 3 (see below).

It is necessary to further constrain $t_{end}$ such that for large TD peaks (*e.g.*, those recorded for SPAN samples) the automatically determined value of $t_{end}$ is at least 10 ds before the upper bound time, $t_n$, of the interval during which the instrument's Hg AF signal is recorded. The VI therefore defines $t_{end}$ as the smaller of the result of eq. 3 and $t_{n-10}$, where $n$ represents the number of AF signal values recorded ($n = 389$ in Fig. 1a and Fig. S4a in the Supporting Information). A minimum value of 10 ds is also prescribed for $t_{end}$ such that for very small TD peaks the automatically determined peak end does not occur at or before

the peak max time (*e.g.*, when $f \geq H_i/A_{SPAN}$ in eq. 3). (See Sect. S3 in the Supporting Information for further details.)

### 2.1.4 Evaluation

I evaluate the performances of the $VI_{m,a}$ and $VI_{a,a}$ methods by applying both methods to laboratory data collected with two Tekran® analyzers (one model 2537A and one model 2537B). One dataset is processed for each analyzer. The test dataset collected with the 2537A instrument is shown in Fig. 2. (The test dataset for the 2537B instrument is shown in Fig. S3.)

I consider manual definition of the Hg thermal desorption peaks (the $VI_{m,m}$ method) to be the benchmark for signal processing accuracy, and I assess the accuracies of the $VI_{m,a}$ and $VI_{a,a}$ methods by comparing Hg sample loadings derived from both methods with loadings derived from the $VI_{m,m}$ method. A similar comparison is used to evaluate the accuracy of the Tekran® method. The performances of all methods are further evaluated and compared based on the Hg limits of detection (LODs) they achieve.



## 3 Results and Discussion

### 3.1 Performance Evaluations

#### 3.1.1 The Tekran® Hg Thermal Desorption Peak Integration Method

For Hg loadings derived from the Tekran® method, $Hg_{Tekran}$, I define absolute bias as $Hg_{Tekran} - Hg_{benchmark}$, where "$Hg_{benchmark}$"

represents loadings derived from the $VI_{m,m}$ method. I define relative bias as $100 \times (Hg_{Tekran} - Hg_{benchmark}) \div Hg_{benchmark}$. As illustrated in Fig. 3 (and Fig. S5 in the Supporting Information), Hg loadings derived using the Tekran® method tend to be biased low, with the relative bias becoming more negative with decreasing loading. I present in Table 1 relative bias values from Fig. 3b at several discrete Hg sample loadings (analogous results for the 2537B dataset are presented in Table S2 in the Supporting Information). The corresponding Hg concentrations ($ng/m^3$) under typical operating conditions for the Tekran®

analyser (5 liter sample volumes) are also shown. My results are consistent with those of Swartzendruber et al. (2009) and Slemr et al. (2016), but also demonstrate that the Tekran® method can produce significant low biases ($\geq 5\%$; see Tables 1 and S2) at tropospheric background GEM and THg concentrations ($\sim$1 to 2 $ng/m^3$; sample loadings of 5−10 pg under typical Tekran® operating conditions).

To further characterize the performance of the Tekran® method, Fig. 4 compares Hg atomic fluorescence baselines calculated

by the Tekran® method and by my VI-based peak height determination methods, for samples with Hg loadings of approximately 0.5, 1, 2, and 4 times the estimated $\sim$0.8 pg LOD achieved with the Tekran® method (see below). (See Sect. S6 of the Supporting Information for details on how I reproduced the baselines calculated by the Tekran® method.) Figure 4 illustrates the tendency of the Tekran® method to truncate the Hg thermal desorption peaks. The peaks tend to become more severely truncated as they become smaller, and as a result, the relative biases in the corresponding Hg loadings tend to become

more negative as the loadings decrease, as shown in Fig. 3b and Fig. S5b in the Supporting Information (see also Fig. 2 in Slemr et al., 2016).

Table 2 presents Hg LODs for the Hg fluorescence signal processing methods and datasets I tested. The nominal Hg limit of detection of the Tekran® analyzer is 0.5 pg (see Sect. S7 of the Supporting Information for further details). However, some Hg thermal desorption peaks in the 2537A dataset are undetected by the Tekran® method for Hg loadings ≤0.8 pg (Figs. 2 and

3a). My results suggest that the actual Hg LOD achieved with the Tekran® method is $\sim$60% higher than the nominal value. By comparison, the LOD achieved with the $VI_{m,m}$ method (estimated as twice the standard deviation of blank samples, $n = 62$) is 0.10 pg.

For the 2537B dataset, all Hg thermal desorption peaks are detected by the Tekran® method for Hg loadings >0.2 pg (Figs. S3 and S5 in the Supporting Information), suggesting the LOD is $\sim$60% lower than the nominal value. The lower Hg LOD

achieved with the Tekran® method when applied to the 2537B dataset is attributable to the hardware modifications that were made to the 2537B instrument to increase its signal-to-noise ratio (Ambrose et al., 2013). The 0.085 pg LOD achieved with the $VI_{m,m}$ method is 56% lower than the Tekran®-based LOD (Table 2).



### 3.1.2 The Automated VI-Based Hg Thermal Desorption Peak Height Determination Method

By comparison with the Tekran® method, the $VI_{a,a}$ method usually identifies the Hg thermal desorption peak baseline with good accuracy, even at Hg loadings near to or below the LOD achieved with the Tekran® method (Fig. 4). As a result, most Hg loadings derived from the $VI_{a,a}$ method are quite accurate (Figs. 2 and 5; Table 1). For samples with Hg loadings below

the estimated 0.8 pg LOD of the Tekran® method, the mean absolute and relative unsigned biases in the loadings derived from the $VI_{a,a}$ method are 0.028±0.005 pg and 15±3%, respectively ($n = 78$). The biases are very small and much smaller in magnitude than those for the Tekran® method (estimated at −0.80 pg and −100%). Based on the equations of the linear regressions shown in Figs. 3a and 5a, the $VI_{a,a}$ method achieves ≥95% reduction in absolute unsigned bias in calculated Hg loadings when compared with the Tekran® method.

The Hg LOD achieved with the $VI_{a,a}$ method is 0.12 pg (estimated as twice the standard deviation of blank values; $n = 62$). This value is 20% higher than LOD achieved with the $VI_{m,m}$ method, but 85% lower than the LOD achieved with the Tekran® method (Table 2). Similarly, the width of the residual distribution in Fig. 5c is 81% narrower than that in Fig. 3c, which reflects the improved analytical precision achieved with the $VI_{a,a}$ method, in comparison with the Tekran® method.

Evaluation of the $VI_{a,a}$ method as in Fig. 5 for the 2537B dataset (Fig. S5, Table S2 in the Supporting Information) yields the

linear regression equation $y = 1.015(2)x − 0.017(8)$ pg ($r^2 = 0.9998$, $n = 132$). For samples with Hg loadings below 0.8 pg, the observed mean absolute and relative unsigned biases in the loadings derived from the $VI_{a,a}$ method are 0.031±0.007 pg and 33±9%, respectively ($n = 41$). The $VI_{a,a}$ method yields larger relative bias when applied to the 2537B dataset than when applied to the 2537A dataset. However, absolute unsigned biases are equivalent (at the 95% confidence interval) for the two datasets. Based on the equations of the linear regressions shown in Figs. S5a and S6a in the Supporting information, the $VI_{a,a}$ method

achieves ≥82% reduction in absolute unsigned bias in calculated Hg loadings when compared with the Tekran® method. The Hg LOD is 0.14 pg (Table 2), which is 59% higher than the value achieved with the $VI_{m,m}$ method, but 31% lower than the value achieved with the Tekran® method. The $VI_{a,a}$ method appears to yield a comparable improvement in analytical precision over the Tekran® method when applied to the 2537A and 2537B datasets (the width of the residual distribution in Fig. S6c is 72% narrower than that in Fig. S5c).

### 25 3.1.3 The Semi-Automated VI-Based Peak Height Determination Method

The $VI_{a,a}$ method poorly identifies the start of the Hg thermal desorption peak for some blank samples with the lowest Hg loadings. As a result, the Hg TD peak height and the calculated Hg loading tend to be underestimated for those samples (Fig. 2; Fig. S3 in the Supporting Information).

I developed the $VI_{m,a}$ method as a compromise between the Hg TD peak definition accuracy achieved with the $VI_{m,m}$ method

and the data processing speed achieved with the $VI_{a,a}$ method. (Peak height determination for a single sample requires from <1 second with the $VI_{a,a}$ method to several seconds for the $VI_{m,m}$ method; the data processing time for the $VI_{m,a}$ method is approximately half that for the $VI_{m,m}$ method.) Comparison of the $VI_{m,a}$ and $VI_{a,a}$ results in Fig. 2 (and Fig. S3 in the Supporting





Information) shows that defining $t_{start}$ manually instead of automatically yields more accurate measurements for samples with the lowest Hg loadings.

Evaluation of the $VI_{m,a}$ method as in Fig. 5 yields the linear regression equation $y = 1.001(1)x + 0.004(3)$ pg ($r^2 = 0.99998$, $n = 152$). Comparison between the latter equation and that of the regression in Fig. 5a suggests that biases in Hg loadings derived from the $VI_{a,a}$ and $VI_{m,a}$ methods are not significantly different over the range of loadings in the 2537A dataset (see also Table 1). However, biases in Hg loadings derived from the $VI_{m,a}$ method are lower at low loadings than those determined for the $VI_{a,a}$ method (Sect. 3.1.2): for samples with Hg loadings below 0.8 pg, the mean absolute and relative unsigned biases in the loadings derived from the $VI_{m,a}$ method are 0.012±0.003 pg and 6±1%, respectively ($n = 78$). The estimated 0.11 pg Hg LOD achieved with the $VI_{m,a}$ method falls in between the values achieved with the $VI_{m,m}$ and $VI_{a,a}$ methods (Table 2).

The $VI_{m,a}$ method performs consistently better for the 2537B dataset than does the $VI_{a,a}$ method. The equation of the linear regression (as in Fig. 5a) is $y = 1.005(1)x + 0.009(5)$ pg ($r^2 = 0.99994$, $n = 132$). For samples with Hg loadings below 0.8 pg, the observed mean absolute and relative unsigned biases in the loadings derived from the $VI_{m,a}$ method are 0.010±0.003 pg and 6±2%, respectively ($n = 41$). The estimated LOD achieved with the $VI_{m,a}$ method is 0.010 pg and falls between the LODs achieved with the $VI_{m,m}$ and $VI_{a,a}$ methods (Table 2).

### 3.1.4. Sensitivity Analyses and Uncertainties

I estimate signal-processing uncertainties in the Hg thermal desorption peak heights derived from the $VI_{m,m}$ method to be equal to twice the mean baseline standard deviation ($\sigma_{bl}$, Sect. 2.1.1), corresponding with a Hg loading of 0.053 pg for the dataset in Fig. 2 (0.030 pg for the dataset in Fig. S3 in the Supporting Information). I estimate signal processing uncertainties in the Hg loadings derived from the Tekran®, $VI_{a,a}$, and $VI_{m,a}$ methods as the sum of the biases in those methods (derived as in Fig. 3) and the resulting increase (relative to the $VI_{m,m}$ method) in the Hg limit of detection (Table 2). For the $VI_{a,a}$ and $VI_{m,a}$ methods, I estimate a conservative threshold uncertainty value of ±2$\sigma_{bl}$.

To test the sensitivity to initialization parameters of Hg loadings derived from the $VI_{a,a}$ and $VI_{m,a}$ methods I recalculated those loadings after making a series of modifications to the method initialization parameters. After each modification I recalculated the parameters of the Hg regression (*e.g.*, Fig. 5a). The modifications tested for the 2537A dataset (described further in Tables S3 and S4 in the Supporting Information) include shifts in $t_{end}$ by ±$\delta t_{end}$ (as defined in Sect. 2.1.3), shifts in $t_{start}$ to the values determined (as described in Sect. 2.1.2) for the second pair of SPAN samples in Fig. 2 (applicable only to the $VI_{a,a}$ method), and initialization of the VI using the second pair of SPAN samples. (Details on the sensitivity tests applied to the 2537B dataset are described in Tables S5 and S6 in the Supporting Information.)

The above modifications result in insignificant changes (at the 95% confidence interval) to the bias parameters derived (as in Fig. 5a) for the $VI_{a,a}$ method and the 2537 dataset (Table S3 in the Supporting Information). Two sensitivity tests applied to the $VI_{a,a}$ method and the 2537B dataset result in significant changes (at the 95% confidence interval) to the calculated bias parameters (Table S5 in the Supporting Information). Results obtained with the $VI_{a,a}$ method therefore appear to be sensitive to initialization parameters in some cases, although signal processing uncertainties remain quite low and well below those




estimated for the Tekran® method (Table 2). It is possible that a large shift in $t_{start}$ for SPAN samples over the course of an analysis would increase the sensitivity of the $VI_{a,a}$ method to initialization parameters. Results obtained with the $VI_{m,a}$ method are insensitive (at the 95% confidence interval) to initialization parameters (Tables S4 and S6 in the Supporting Information). I estimate that the overall "signal processing" uncertainty attributable to my $VI_{a,a}$ method, as applied to the 2537A dataset, is

within ±[0.2% + 0.053 pg]. (The first term in the latter expression represents the slope of the regression in Fig. 5a; the second term represents $2\sigma_{bl}$, which in this case is larger than the sum of the 0.000±0.006 pg intercept of the fit in Fig. 5a and the 0.02 pg difference in Hg LODs achieved with the $VI_{a,a}$ and $VI_{m,m}$ methods). I estimate that overall signal processing uncertainty attributable to the $VI_{m,a}$ method is also within ±[0.2% + 0.053 pg]. By comparison, overall signal processing uncertainty attributable to the Tekran® method (considering bias quantified in Sect. 3.1.1 and the estimated Hg LOD) is within ±[4% +

0.94 pg]. The above uncertainty results, together with estimated Hg LODs and analogous results for the 2537B dataset, are summarized in Table 2. For both test datasets, signal processing uncertainties estimated for my VI-based methods are significantly lower than those for the Tekran® method. Signal processing uncertainty ranks as follows for the VI-based methods: $VI_{m,m} \leq VI_{m,a} < VI_{a,a}$.

## 4. Conclusions and Implications

I describe three improved methods for processing the raw Hg atomic fluorescence signal from Tekran® 2537A and 2537B Hg vapor analyzers. The methods incorporate manual, semi-automated, or fully automated Hg thermal desorption peak identification processes. I implement my methods through a Virtual Instrument in National Instruments™ LabVIEW and evaluate them, together with the Tekran® internal Hg TD peak integration method, using test datasets from two Tekran® instruments (one 2537A and one 2537B).

Consistent with previous work (Swartzendruber et al., 2009; Slemr et al., 2016), my results demonstrate that Hg loadings derived from the Tekran® method tend to be biased low, with the relative bias becoming more negative with decreasing loading. It follows that the magnitude of the bias in Tekran®-based Hg measurements will depend significantly on sampling conditions (*e.g.*, sample concentration and volume). Therefore, I recommend that signal processing bias be examined, and associated uncertainties be quantified, in all future applications of the Tekran® instruments, regardless of sampling arrangement.

With respect to atmospheric GEM and THg measurements, my results demonstrate that the Tekran® method can produce significant low biases (≥5%) at background concentrations (~1 to 2 ng/m³) under typical operating conditions (Hg loadings of 5−10 pg). My results therefore indicate that post-processing the raw Tekran® data will yield significant improvements in the accuracy of the derived Hg concentrations under much broader environmental conditions than previously recognized. Such conditions should not be assumed to be limited to those where GEM concentrations can be significantly depleted, as can occur

in the free troposphere and lower stratosphere (*e.g.*, Talbot et al., 2007; Lyman and Jaffe, 2012; Timonen et al., 2012; Gratz et al., 2015), and at the surface in polar and mid-latitude regions under special photochemical conditions (*e.g.*, Schroeder et al., 1998; Obrist et al., 2011).



Under typical sampling conditions employed with the Tekran® 1130/1135 Hg speciation system (Landis et al., 2002), with sample collection volumes on the order of 600−1,800 standard liters, Hg loadings during GOM and PBM analyses will be low (<5 pg) for ambient concentrations <8 pg/m$^3$. Relative biases much greater than 5% in Tekran®-based GOM and PBM measurements should therefore be limited to the lowest measured ambient concentrations. The distribution of GOM or PBM

reduction over multiple denuder or particulate filter desorption cycles, respectively, would tend to reduce effective sample loadings and therefore exacerbate signal processing bias in applications of the Tekran® speciation system.

My results demonstrate that signal processing uncertainties may represent a significant fraction of the overall uncertainty in Tekran®-based atmospheric Hg measurements. I estimate that signal processing uncertainties in Hg loadings derived from the Tekran® method are within ±[4% + 0.94 pg] for the 2537A dataset and within ±[6% + 0.21 pg] for the 2537B dataset. By

comparison, non-signal processing related uncertainties in Tekran®-based GEM, TGM, and THg measurements were previously estimated to be on the order of ~10% to 15% (Ambrose et al., 2011; Ambrose et al., 2013, 2015). Quantification of signal processing uncertainties in future applications of the Tekran® 2537 instrument will substantially improve the quality of the resulting measurements.

My results suggest that the performance of the Tekran® method may be improved with hardware modifications that increase

the instrument's signal-to-noise ratio. The Tekran® method performed slightly better when applied to the 2537B dataset than when applied to the 2537A dataset, particularly at low Hg loading. The primary difference between the 2537A and 2537B instruments I tested is that the 2537B instrument was modified to improve its signal-to-noise ratio by replacing the standard sample cuvette and detector bandpass filter with a mirrored cuvette and an improved filter, respectively (Ambrose et al., 2013). The cuvette and bandpass filter were obtained from Tekran® Instruments (Part Numbers 40-25105-00 and 40-25200-02,

respectively). It is possible that the performance of the Tekran® method can also be improved through modification of the method's integration parameters, though I tested only the default parameters (Table S1).

Estimated signal processing uncertainties in Hg loadings derived from my methods range from within ±0.053 pg when the Hg thermal desorption peaks are defined manually to within ±[0.2% + 0.053 pg] (2537A dataset) and ±[2% + 0.080 pg] (2537B dataset) when Hg TD peak definition is fully automated. Biases in Hg loadings derived from my methods are lower by >80%

than biases derived from the Tekran® method. Limits of detection for Hg decrease by 31−88% when my methods are used in place of the Tekran® method.

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

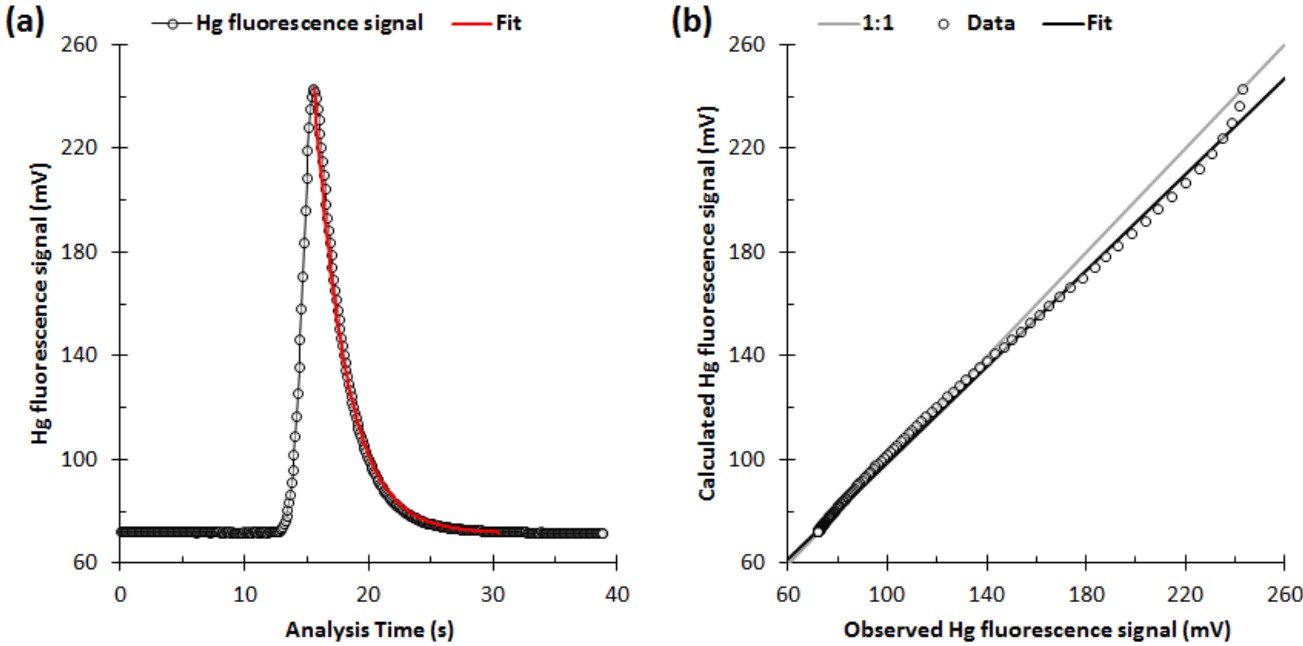

**Figure 1: (a) Example Hg thermal desorption profile during a calibration gas analysis cycle on a Tekran® 2537A instrument. Also shown is the corresponding 150 point exponential Bisquare (unweighted) regression fit (eq. 1; $r^2 = 0.998$) used to derive the decay constant ($b = -0.041 \pm 0.004$ ds$^{-1}$) during initialization of the VI's signal processing method. (b) Comparison between the calculated (fit) and observed Hg atomic fluorescence signal values in panel (a). The slope and intercept of the linear fit are $0.921 \pm 0.007$ and $7.2 \pm 0.8$ mV, respectively (uncertainties are 95% confidence intervals). (Analogous results obtained with a 2537B analyzer are**
**shown in Fig. S4 in the Supporting Information.)**



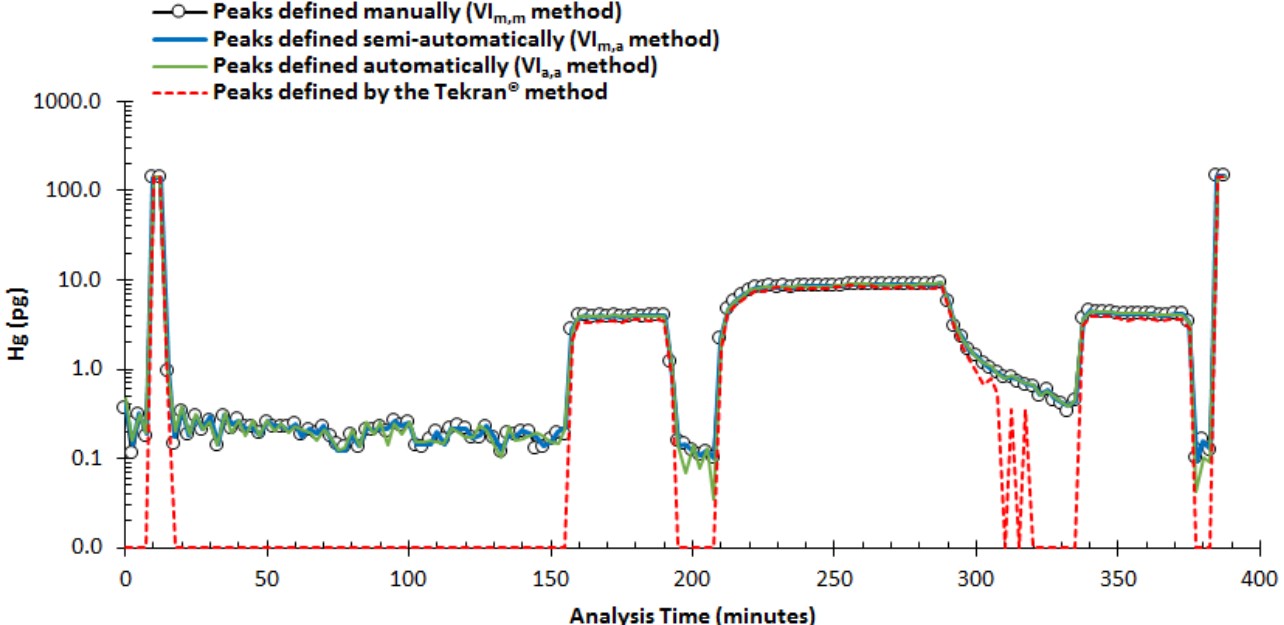

**Figure 2: Test dataset collected with a Tekran® 2537A instrument, represented as Hg loadings derived from my VI-based manual, semi-automated, and automated Hg thermal desorption peak height determination methods (the $VI_{m,m}$, $VI_{m,a}$, and $VI_{a,a}$ methods, respectively), and by the Tekran® method. Peaks not detected by the Tekran® method are assigned a value of 0.01 pg. The two pairs of data points at >100 pg correspond with calibration gas analysis cycles (SPAN samples). I use the first pair of SPAN samples to initialize the VI (as described in Sect. 2.1.2). I use response factors calculated from the second pair of SPAN samples (and the preceding pair of blanks) to calculate Hg loadings for all other samples in the dataset. The mean value of the baseline standard deviation, $\sigma_{bl}$ (defined in Sect. 2.1.1), is ~0.03 mV (equal to ~0.03 pg). The corresponding estimated lower-limit $f$ value (mean $\pm 2\sigma$; Sect. 2.1.3) is $1.86(1) \times 10^{-4}$.**





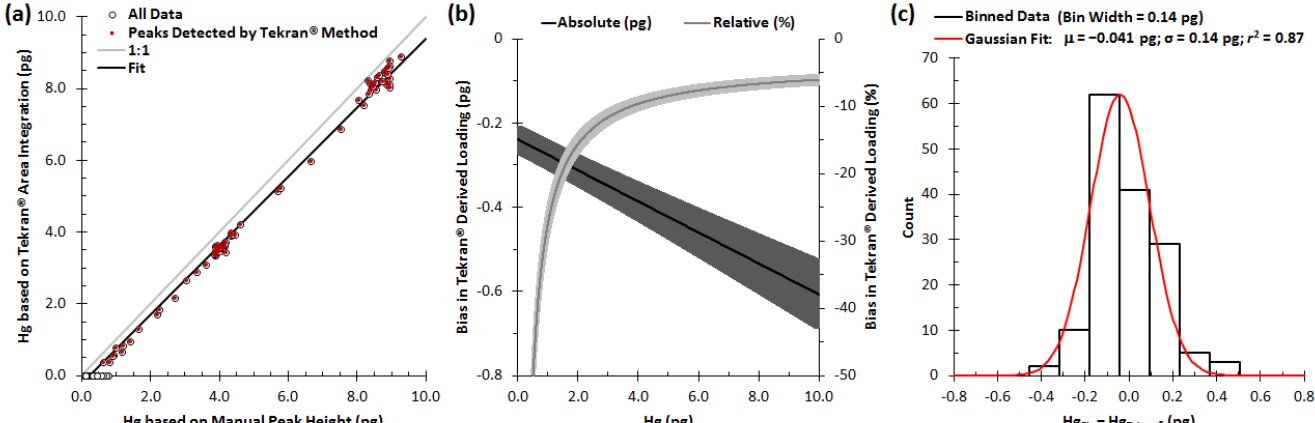

**Figure 3: (a) Comparison of Hg loadings derived from measurements made with a Tekran® 2537A instrument using the Tekran® method and my VI-based Hg thermal desorption peak height determination method (dataset shown in Fig. 2, excluding SPANs), with the peaks defined manually (the VI$_{m,m}$ method). The equation of the linear regression is $y = 0.963(7)x - 0.24(3)$ pg ($r^2 = 0.998$, $n = 152$). The fit excludes data derived from peaks not detected by the Tekran® method (represented by the open symbols). (b) Absolute and relative biases in the Tekran® derived Hg loadings, based on the fit in panel (a). The grey bands represent propagated uncertainties (95% confidence intervals) in the fit parameters. (c) Distribution of residuals from panel (a), including only data derived from detected peaks. (Analogous results for the 2537B instrument are presented in Fig. S5 in the Supporting Information.)**





**Figure 4: Comparison of Hg fluorescence baselines calculated by applying manual (VI$_{m,m}$ method), automatic (Tekran® and VI$_{a,a}$ methods), and semi-automatic (VI$_{m,a}$ method) Hg thermal desorption peak definition methods to samples with Hg loadings of approximately (a) 0.5, (b) 1, (c) 2, and (d) 4 times the limit of detection of the Tekran® method (~0.8 pg). Measurements were made with a 2537A instrument. Biases in the Hg loadings derived from the Tekran® method and the VI-based methods are indicated. Biases are expressed relative to the loadings derived from the VI$_{m,m}$ method and are negative when the VI$_{m,m}$ based loadings are higher. The Tekran® baselines are missing from panels (a) and (b) because the peaks were not detected by the Tekran® method.**





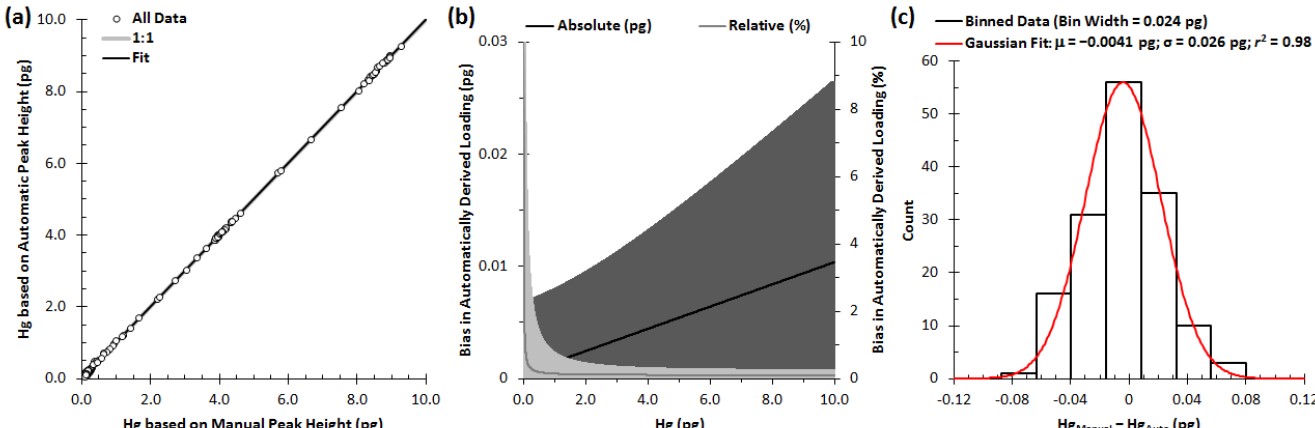

**Figure 5: (a) Comparison of Hg loadings derived from measurements made with a Tekran® 2537A instrument using my VI-based automated and manual peak height determination methods (the VI$_{a,a}$ and VI$_{m,m}$ methods, respectively; dataset shown in Fig. 2, excluding SPANs). The equation of the linear regression is $y = 1.001(1)x + 0.000(6)$ pg ($r^2 = 0.99992$, $n = 152$). (b) Absolute and relative biases in the VI-based Hg loadings, based on the fit in panel (a); grey bands represent propagated uncertainties (95% confidence intervals) in the parameters of the fit in panel (a). (c) Distribution of residuals from panel (a).**

**Table 1. Bias in Hg loadings derived by applying automated and semi-automated Hg atomic fluorescence signal processing methods to measurements made with a Tekran® 2537A instrument.**

| Method | Hg (pg, ng/m³)[a,b] | | | | | | |
|---|---|---|---|---|---|---|---|
| | 7.5, 1.5 | 3.75, 0.75 | 2.5, 0.5 | 1,25, 0.25 | 0.5, 0.1 | 0.25, 0.05 | 0.125, 0.025 |
| | Bias (%) | | | | | | |
| Tekran®[c] | −7.4 ± 0.9 | −11 ± 1 | −14 ± 1 | −23 ± 3 | −100[f] | −100[f] | −100[f] |
| VI$_{a,a}$[d] | 0.1 ± 0.2 | 0.1 ± 0.2 | 0.1 ± 0.3 | 0.2 ± 0.5 | 0.3 ± 1.2 | 0.6 ± 2.3 | 1 ± 5 |
| VI$_{m,a}$[e] | 0.15 ± 0.08 | 0.2 ± 0.1 | 0.3 ± 0.1 | 0.5 ± 0.3 | 1.0 ± 0.6 | 2 ± 1 | 4 ± 2 |

[a]Bias values for the Tekran® and VI$_{a,a}$ methods are calculated from the equations of the linear regressions in Figs. 3a and 5a, respectively. Bias values for the VI$_{m,a}$ method are similarly calculated from the linear regression equation given in Table S4 in the Supporting Information ("Standard" configuration). All bias values are expressed relative to Hg loadings derived by processing the data using manual peak definition (the VI$_{m,m}$ method).
[b]Hg loadings are also expressed in terms of concentrations under the typical Tekran® operating parameters.
[c]Tekran® operating and peak integration parameters are defined in Table S1.
[d]My VI-based peak height determination method, with peak start and end times determined automatically (VI$_{a,a}$).
[e]My VI-based peak height determination method, with peak start times determined manually and peak end times determined automatically (VI$_{m,a}$).
[f]For Hg < the estimated 0.8 pg Tekran® limit of detection, the true bias is −100%. For clarity, the true bias is substituted for the calculated values.




**Table 2. Estimated signal processing uncertainties and Hg limits of detection (LODs) achieved with the Tekran® peak integration method and my VI-based peak height determination methods as applied to measurements made with Tekran® 2537A and 2537B instruments.**

| Method[a] | Signal Processing Uncertainty[b] | Hg LOD (pg) |
|---|---|---|
| *2537A Dataset[c]* | | |
| Tekran® | ±[4% + 0.94 pg] | 0.80[d] |
| $VI_{m,m}$ | ±0.053 pg | 0.10[e] |
| $VI_{a,a}$ | ±[0.2% + 0.053 pg] | 0.12[e] |
| $VI_{m,a}$ | ±[0.2% + 0.053 pg] | 0.10[e] |
| *2537B Dataset[f]* | | |
| Tekran® | ±[6% + 0.21 pg] | 0.20[d] |
| $VI_{m,m}$ | ±0.030 pg | 0.085[g] |
| $VI_{a,a}$ | ±[2% + 0.080 pg] | 0.13[g] |
| $VI_{m,a}$ | ±[0.6% + 0.030 pg] | 0.10[g] |

[a]The "Tekran®" method is the Tekran® analyzer's internal automated Hg thermal desorption peak integration method, parameterized as indicated in Table S1 in the Supporting Information. The "$VI_{m,m}$", "$VI_{a,a}$", and "$VI_{m,a}$" Hg TD peak height determination methods were developed in this work and are described in Sect. 2.1.2. The operating parameters of the Tekran® analyzers are presented in Table S1.
[b]Estimated as described in Sect. 3.1.4.
[c]The 2537A dataset is shown in Fig. 2.
[d]Estimated as the highest Hg loading derived from the $VI_{m,m}$ method for samples for which the Tekran® method failed to detect the Hg TD peak.
[e]Estimated as twice the standard deviation of blank loadings ($n = 62$).
[f]The 2537B dataset is shown in Fig. S3 in the Supporting Information.
[g]Estimated as twice the standard deviation of blank loadings ($n = 37$).