# Peer review of "Improved Methods for Signal Processing in Measurements of Mercury by Tekran® 2537A and 2537B Instruments"

_Atmospheric Measurement Techniques, 2017_

## Referee Comment (RC1) · F. Slemr (Referee) · 24 Aug 2017

The author reports a systematic bias in Hg measurements caused by default signal integration of Tekran instruments. This matters since these instruments are widespread and used not only for a measurement of elemental mercury, as mentioned in the title of the article, but they also serve as detectors for different mercury speciation methods. The bias has been reported before but its wide reaching consequences have mostly been neglected so far. The paper will hopefully help to make the mercury community aware of this problem and contribute to the improvement of accuracy and precision of worldwide measurements of mercury and its species.

[Figure]

The methods of signal processing are clearly described. The paper is well organised and written. It should be published with small changes suggested below.

Comments

Title: The implications of the improved processing of Tekran signal are even more important for measurements of gaseous oxidized mercury (GOM) and particulate bond mercury (PBM) than for measurements gaseous elemental mercury (GEM). A title without mentioning specifically "elemental mercury" would thus be more appropriate.

The author claims that the Tekran measurements even with loads $\geq$ 10 pg are biased low. This is probably not correct because it will depend on the calibration which uses the same default integration. If the calibration is made with 10 pg than then the measurement of 10 pg, both with the same default integration, will not be biased (the low bias of the calibration equals the low bias of the measurement and they cancel out). This should be mentioned and the text modified accordingly.

Page 2, line 23: A sentence describing the hardware change of the 2537B instrument would be helpful.

Page 5, lines 11-14: Three consecutive sentences start with "I estimate the uncertainty..."

Page 10, 1st paragraph: This paragraph concerning the GOM and PBM measurements with the Tekran speciation system is confusing and grossly underrates the problem. GOM and PBM measurements in AMNet could be mentioned as an example: With 2 h sampling at a rate of 10 l min-1 median GOM concentrations of 1.2 – 2.5 pg m-3 (Gay et al., ACP, 13, 11339-11349, 2013) will give loads of 1.4 – 3 pg. I.e. half of these measurements are made with loads below 3 pg (not to say anything about 8 pg mentioned in this paragraph as a minimum load to avoid large bias) and are thus substantially biased low with correspondingly poor precision. PBM measurements and other networks could be mentioned as well. As most of the existing GOM and PBM data

have been obtained by Tekran speciation system it can be said, without exaggeration, that most of them are biased low. The bias adds to the intensively discussed artefacts related to the GOM and PBM sampling.

Page 10, paragraph starting at line 14: Longer sampling times are another (hardware) way to increase the loads and thus reduce the biases and improve the precision especially of the GEM and TGM measurements. E.g. GEM measurements at stations in the southern hemisphere such as at Cape Point and Amsterdam Island are for these reasons already being made with 15 min sampling time which at a sampling rate of 1 l min-1 yields loads of ∼15 pg. For GOM and PBM measurements longer sampling times may not be feasible because of sampling artefacts and poor temporal resolution.

Fig. 2: There is hardly any contrast between the colours for methods VIm,a and VIa,a.

---

## Referee Comment (RC2) · Anonymous Referee #3 · 26 Oct 2017

Supports previous work identifying the need for external peak integration at low load levels. Manually identification of start and/or end peaks is impractical and could be biased. Use automated (Vaa) identification Uncertain why Tekran LOD's were calculated using 2.5 liter sample volumes when most sample 5 liter volumes.

---

## Author Comment (AC1) · 26 Oct 2017

The reviewer's comments are shown below, followed in sequence by my responses.

Comment 1: Title: The implications of the improved processing of Tekran signal are even more important for measurements of gaseous oxidized mercury (GOM) and particulate bond mercury (PBM) than for measurements gaseous elemental mercury (GEM). A title without mentioning specifically "elemental mercury" would thus be more appropriate.

Response: I agree that a title without mentioning specifically "elemental mercury"

[Figure]

would be more appropriate. I changed the title accordingly, omitting "Elemental" and "Vapor".

Comment 2: The author claims that the Tekran measurements even with loads $\geq$ 10 pg are biased low. This is probably not correct because it will depend on the calibration which uses the same default integration. If the calibration is made with 10 pg than then the measurement of 10 pg, both with the same default integration, will not be biased (the low bias of the calibration equals the low bias of the measurement and they cancel out). This should be mentioned and the text modified accordingly.

Response: The reviewer makes a very good point that I agree should be addressed in the paper. I changed the text in several places to clarify that the Tekran method "can" yield Hg measurements that are biased low at high loading ($\geq$ 10 pg). For instance, line 13 of Page 1 now reads "the Tekran® method can produce significant low biases", not "the Tekran method produces significant low biases". Additionally, I added the following paragraph to Sect. 3.1.1.:

"It is possible that bias introduced by the Tekran® method can be made smaller by calibrating at loadings more similar to the loadings in the samples of interest. The measurements in Figs. 3a and S5a are calibrated with SPAN loadings >10 times higher (see Figs. 2 and S3, respectively). Calibrating the measurements in Fig. S5a using the external SPANS (Fig. S3; loadings >5 times higher) yields the linear regression equation (as in Fig. S5a) y = 0.96(1)x − 0.09(5) pg. The slope of the latter equation is closer to unity (though not significantly) than that of the equation derived from Fig. S5a."

I added the following sentences to Sect. 4: "Additionally, it is possible that measurement bias introduced by the Tekran® method can be made smaller by calibrating at loadings more similar to the loadings in the samples of interest. My results demonstrate a minor reduction in bias when measurements made with the 2537B instrument are calibrated at loadings that are >5 times higher rather than at loadings >10 times

higher."

Comment 3: Page 2, line 23: A sentence describing the hardware change of the 2537B instrument would be helpful.

Response: I agree. I added the following brief description of the hardware changes to Sect. 2, including the manufacturer's part numbers for the components that were changed:

"The instrument's sample cuvette and detector bandpass filter were replaced with a mirrored cuvette and an improved bandpass filter, respectively (Ambrose et al., 2013). The cuvette and bandpass filter were obtained from Tekran® (Part Numbers 40-25105-00 and 40-25200-02, respectively)."

Comment 4: Page 5, lines 11-14: Three consecutive sentences start with "I estimate the uncertainty. . ."

Comment: Trying to somehow combine these sentences seems to complicate them. However, I did reword the latter two sentences to avoid some redundancy.

Comment 5: Page 10, 1st paragraph: This paragraph concerning the GOM and PBM measurements with the Tekran speciation system is confusing and grossly underrates the problem. GOM and PBM measurements in AMNet could be mentioned as an example: With 2 h sampling at a rate of 10 l min-1 median GOM concentrations of 1.2 – 2.5 pg m-3 (Gay et al., ACP, 13, 11339-11349, 2013) will give loads of 1.4 – 3 pg. I.e. half of these measurements are made with loads below 3 pg (not to say anything about 8 pg mentioned in this paragraph as a minimum load to avoid large bias) and are thus substantially biased low with correspondingly poor precision. PBM measurements and other networks could be mentioned as well. As most of the existing GOM and PBM data have been obtained by Tekran speciation system it can be said, without exaggeration, that most of them are biased low. The bias adds to the intensively discussed artefacts related to the GOM and PBM sampling.

Response: I agree that signal-processing bias/uncertainties in Tekran[®]-based GOM and PBM measurements can be even more severe than in GEM/THg measurements under common sampling conditions and that the text should be modified to more accurately reflect the severity of the issue. I removed the paragraph in question and replaced it with the following one:

"Most measurements of atmospheric GOM and PBM made with the Tekran[®] 1130/1135 Hg speciation system (Landis et al., 2002) may be significantly biased low. For instance, "typical" median GOM and PBM concentrations measured at 21 Atmospheric Mercury Network (AMNet) sites in the United States and Canada during the years 2009–2011 were in the ranges 1.2–2.5 pg/m3 and 2.5–5.0 pg/m3, respectively. The corresponding Hg loadings are 1.4–3.0 pg and 3.0–6.0 pg. Median signal processing bias, estimated as $100 \times$ (HgTekran – Hgbenchmark) Ãǔ HgTekran, would be within −39% and –19%, respectively, based on the fit in Fig. 3a and within −17% and −12%, respectively, based on the fit in Fig. S5a. Similarly, median concentrations of GOM and/or PBM measured at 10 sites in Canada during the years 2002–2011 were typically <5 pg/m3 (Cole et al., 2014), corresponding with sample loadings of 3–9 pg. The corresponding median signal processing bias would be within −19% and –8%, respectively, based on the fit in Fig. 3a and within −12% and –8%, respectively, based on the fit in Fig. S5a."

Additionally, I added the following sentence to the Abstract: "Signal processing bias can also add significantly to uncertainties in Tekran[®]-based gaseous oxidized mercury (GOM) and particle bound mercury (PBM) measurements, which often derive from Hg sample loadings <5 pg." I feel that a more comprehensive review of previous GOM and PBM measurements is beyond the scope of this paper.

Comment 6: Page 10, paragraph starting at line 14: Longer sampling times are another (hardware) way to increase the loads and thus reduce the biases and improve the precision especially of the GEM and TGM measurements. E.g. GEM measurements at

stations in the southern hemisphere such as at Cape Point and Amsterdam Island are for these reasons already being made with 15 min sampling time which at a sampling rate of 1 l min-1 yields loads of âĹij15 pg. For GOM and PBM measurements longer sampling times may not be feasible because of sampling artefacts and poor temporal resolution.

Response: I agree and added the following sentence to Sect. 4: "Measurement bias and precision can in principle both be improved by employing longer sample pre-concentration times and/or higher sample flow rates to achieve higher sample load-ings." I feel it is beyond the scope of this paper to attempt to address all of the po-tential drawbacks to operating the Tekran$^{®}$ instrument with longer-than-normal pre-concentration times and/or higher-than-normal sample flow rates.

Comment 7: Fig. 2: There is hardly any contrast between the colours for methods VIm,a and VIa,a.

Response: I changed the color of the VIa,a series to improve the contrast against the VIm,a series in Figs. 2 and S3.

Corrections:

I found that the fit in Fig. 3a in the main manuscript included the data points for ther-mal desorption peaks not detected by the Tekran$^{®}$ method. Those points should be excluded. The fit parameters do differ depending on whether those points are included in the fit. For instance, when those points are included, the equation of the fit in Fig. 3a is y = 0.963(7) − 0.24(3) pg (n = 152, r2 = 0.998). When those points are excluded, the fit equation is y = 1.00(1) − 0.45(8) pg (n = 74, r2 = 0.997). This change does not substantially influence my conclusions.

In the revised manuscript, the fit in Fig. 3a excludes data points for TD peaks unde-tected by the Tekran$^{®}$ method. Accordingly, I revised Fig. 3b, Fig. 3c, Tekran$^{®}$ bias values in Table 1, Tekran$^{®}$ signal processing uncertainty values in Table 2, and other

affected numerical values in the text.

---

## Author Comment (AC2) · 26 Oct 2017

Although the reviewer recommended that the manuscript be published "as is", I am providing responses the reviewer's comments here. The reviewer's comments are shown below, followed in sequence by my responses.

Comment 1: Supports previous work identifying the need for external peak integration at low load levels. Manually identification of start and/or end peaks is impractical and could be biased. Use automated (Vaa) identification

Response: By "impractical", I assume the reviewer is referring to the extra time needed

to carry out manual peak identification, as opposed to automated peak identification. Automatic peak identification is certainly the most time efficient, but the performance of any automated method must be evaluated as I have done. Regarding bias, I would argue that it can be minimized with training. It is important to treat all peaks consistently so that any relative bias cancels out during calibration.

Comment 2: Uncertain why Tekran LOD's were calculated using 2.5 liter sample volumes when most sample 5 liter volumes.

Response: The mass-based LODs I calculated are independent of sample volume. At a given Hg concentration, higher sample volumes will be required to achieve higher Hg loadings.

———————————————————

---

## Author Response (AR1)

**Response to Comments by Franz Slemr (Referee #1)**

The reviewer's comments are shown below in bold type.  My responses are in plain type.

**Title: The implications of the improved processing of Tekran signal are even more important for measurements of gaseous oxidized mercury (GOM) and particulate bond mercury (PBM) than for measurements gaseous elemental mercury (GEM). A title without mentioning specifically "elemental mercury" would thus be more appropriate.**

I agree that a title without mentioning specifically "elemental mercury" would be more appropriate.  I changed the title accordingly, omitting "Elemental" and "Vapor".

**The author claims that the Tekran measurements even with loads ≥ 10 pg are biased low. This is probably not correct because it will depend on the calibration which uses the same default integration. If the calibration is made with 10 pg than then the measurement of 10 pg, both with the same default integration, will not be biased (the low bias of the calibration equals the low bias of the measurement and they cancel out). This should be mentioned and the text modified accordingly.**

The reviewer makes a very good point that I agree should be addressed in the paper.  I changed the text in several places to clarify that the Tekran method "can" yield Hg measurements that are biased low at high loading (≥ 10 pg).  For instance, line 13 of Page 1 now reads "the Tekran® method can produce significant low biases", not "the Tekran method produces significant low biases".  Additionally, I added the following paragraph to Sect. 3.1.1.:

"It is possible that bias introduced by the Tekran® method can be made smaller by calibrating at loadings more similar to the loadings in the samples of interest.  The measurements in Figs. 3a and S5a are calibrated with SPAN loadings >10 times higher (see Figs. 2 and S3, respectively).  Calibrating the measurements in Fig. S5a using the external SPANS (Fig. S3; loadings >5 times higher) yields the linear regression equation (as in Fig. S5a) $y = 0.96(1)x - 0.09(5)$ pg.  The slope of the latter equation is closer to unity (though not significantly) than that of the equation derived from Fig. S5a."

I added the following sentences to Sect. 4: "Additionally, it is possible that measurement bias introduced by the Tekran® method can be made smaller by calibrating at loadings more similar to the loadings in the samples of interest.  My results demonstrate a minor reduction in bias when measurements made with the 2537B instrument are calibrated at loadings that are >5 times higher rather than at loadings >10 times higher."

**Page 2, line 23: A sentence describing the hardware change of the 2537B instrument would be helpful.**

I agree.  I added the following brief description of the hardware changes to Sect. 2, including the manufacturer's part numbers for the components that were changed: "The instrument's sample cuvette and detector bandpass filter were replaced with a mirrored cuvette and an improved bandpass filter, respectively (Ambrose et al., 2013).  The cuvette and bandpass filter were obtained from Tekran® (Part Numbers 40-25105-00 and 40-25200-02, respectively)."

**Page 5, lines 11-14: Three consecutive sentences start with "I estimate the uncertainty. . ."**

Trying to somehow combine these sentences seems to complicate them.  However, I did reword the latter two sentences to avoid some redundancy.

**Page 10, 1st paragraph: This paragraph concerning the GOM and PBM measurements with the Tekran speciation system is confusing and grossly underrates the problem. GOM and PBM measurements in AMNet could be mentioned as an example: With 2 h sampling at a rate of 10 l min-1 median GOM concentrations of 1.2 – 2.5 pg m-3 (Gay et al., ACP, 13, 11339-11349, 2013) will give loads of 1.4 – 3 pg. I.e. half of these measurements are made with loads below 3 pg (not to say anything about 8 pg mentioned in this paragraph as a minimum load to avoid large bias) and are thus substantially biased low with correspondingly poor precision. PBM measurements and other networks could be mentioned as well. As most of the existing GOM and PBM data have been obtained by Tekran speciation system it can be said, without exaggeration, that most of them are biased low. The bias adds to the intensively discussed artefacts related to the GOM and PBM sampling.**

I agree that signal-processing bias/uncertainties in Tekran®-based GOM and PBM measurements can be even more severe than in GEM/THg measurements under common sampling conditions and that the text should be modified to more accurately reflect the severity of the issue.  I removed the paragraph in question and replaced it with the following one:

"Most measurements of atmospheric GOM and PBM made with the Tekran® 1130/1135 Hg speciation system (Landis et al., 2002) may be significantly biased low.  For instance, "typical" median GOM and PBM concentrations measured at 21 Atmospheric Mercury Network (AMNet) sites in the United States and Canada during the years 2009–2011 were in the ranges 1.2–2.5 pg/m$^3$ and 2.5–5.0 pg/m$^3$, respectively.  The corresponding Hg loadings are 1.4–3.0 pg and 3.0–6.0 pg.  Median signal processing bias, estimated as 100 × ($Hg_{Tekran}$ – $Hg_{benchmark}$) ÷ $Hg_{Tekran}$, would be within –39% and –19%, respectively, based on the fit in Fig. 3a and within –17% and –12%, respectively, based on the fit in Fig. S5a.  Similarly, median concentrations of GOM and/or PBM measured at 10 sites in Canada during the years 2002–2011 were typically <5 pg/m$^3$ (Cole et al., 2014), corresponding with sample loadings of 3–9 pg.  The corresponding median signal processing bias would be within –19% and –8%, respectively, based on the fit in Fig. 3a and within –12% and –8%, respectively, based on the fit in Fig. S5a."

Additionally, I added the following sentence to the Abstract: "Signal processing bias can also add significantly to uncertainties in Tekran®-based gaseous oxidized mercury (GOM) and particle bound mercury (PBM) measurements, which often derive from Hg sample loadings <5 pg."  I feel that a more comprehensive review of previous GOM and PBM measurements is beyond the scope of this paper.

**Page 10, paragraph starting at line 14: Longer sampling times are another (hardware) way to increase the loads and thus reduce the biases and improve the precision especially of the GEM and TGM measurements. E.g. GEM measurements at stations in the southern hemisphere such as at Cape Point and Amsterdam Island are for these reasons already being made with 15 min sampling time which at a sampling rate of 1 l min-1**

**yields loads of ~15 pg. For GOM and PBM measurements longer sampling times may not be feasible because of sampling artefacts and poor temporal resolution.**

I agree and added the following sentence to Sect. 4: "Measurement bias and precision can in principle both be improved by employing longer sample pre-concentration times and/or higher sample flow rates to achieve
5   higher sample loadings." I feel it is beyond the scope of this paper to attempt to address all of the potential drawbacks to operating the Tekran® instrument with longer-than-normal pre-concentration times and/or higher-than-normal sample flow rates.

**Fig. 2: There is hardly any contrast between the colours for methods VIm,a and VIa,a.**

I changed the color of the $VI_{a,a}$ series to improve the contrast against the $VI_{m,a}$ series in Figs. 2 and S3.

10   Corrections:

I found that the fit in Fig. 3a in the main manuscript included the data points for thermal desorption peaks not detected by the Tekran® method. Those points should be excluded. The fit parameters do differ depending on whether those points are included in the fit. For instance, when those points are included, the equation of the fit in Fig. 3a is $y = 0.963(7) - 0.24(3)$ pg ($n = 152$, $r^2 = 0.998$). When those points are excluded, the fit equation is
15   $y = 1.00(1) - 0.45(8)$ pg ($n = 74$, $r^2 = 0.997$). This change does not substantially influence my conclusions.

In the revised manuscript, the fit in Fig. 3a excludes data points for TD peaks undetected by the Tekran® method. Accordingly, I revised Fig. 3b, Fig. 3c, Tekran® bias values in Table 1, Tekran® signal processing uncertainty values in Table 2, and other affected numerical values in the text.

30

**Response to Comments by Anonymous Referee #3**

Although the reviewer recommended that the manuscript be published "as is", I am providing responses the reviewer's comments here.  The reviewer's comments are shown below in bold type.  My responses are in plain type.

5 **Supports previous work identifying the need for external peak integration at low load levels. Manually identification of start and/or end peaks is impractical and could be biased. Use automated (Vaa) identification**

By "impractical", I assume the reviewer is referring to the extra time needed to carry out manual peak identification, as opposed to automated peak identification.  Automatic peak identification is certainly the most
10 time efficient, but the performance of any automated method must be evaluated as I have done.  Regarding bias, I would argue that it can be minimized with training.  It is important to treat all peaks consistently so that any relative bias cancels out during calibration.

**Uncertain why Tekran LOD's were calculated using 2.5 liter sample volumes when most sample 5 liter volumes.**

15 The mass-based LODs I calculated are independent of sample volume.  At a given Hg concentration, higher sample volumes will be required to achieve higher Hg loadings.

[revised manuscript text omitted]